# Biomarkers of Immunotherapy Response in Patients with Non-Small-Cell Lung Cancer: Microbiota Composition, Short-Chain Fatty Acids, and Intestinal Permeability

**DOI:** 10.3390/cancers16061144

**Published:** 2024-03-13

**Authors:** Alba Moratiel-Pellitero, María Zapata-García, Marta Gascón-Ruiz, Andrea Sesma, Elisa Quílez, Ariel Ramirez-Labrada, Luis Martínez-Lostao, María Pilar Domingo, Patricia Esteban, Alfonso Yubero, Raquel Barbero-Herranz, Ana Moreno-Blanco, José Ramón Paño, Rodrigo Lastra, Julián Pardo, Dolores Isla, Rosa del Campo, Eva Gálvez

**Affiliations:** 1Medical Oncology Department, University Hospital Lozano Blesa, Institute for Health Research Aragón, 50009 Zaragoza, Spain; amoratielp@salud.aragon.es (A.M.-P.); mzapatag@salud.aragon.es (M.Z.-G.); equilezb@salud.aragon.es (E.Q.); ayuberoe@salud.aragon.es (A.Y.); rlastrad@salud.aragon.es (R.L.); disla@salud.aragon.es (D.I.); 2Medical Oncology Department, Miguel Servet University Hospital, 50009 Zaragoza, Spain; mgasconr@salud.aragon.es (M.G.-R.); asesmag@salud.aragon.es (A.S.); 3Aragon Health Research Institute, 50018 Zaragoza, Spain; aramirez@iisaragon.es (A.R.-L.); lmartinezlos@salud.aragon.es (L.M.-L.); pesteban@iisaragon.es (P.E.); pardojim@unizar.es (J.P.); 4Center for Biomedical Research in the Network of Infectious Diseases (CIBERINFEC), Carlos III Health Institute (ISCIII), 28040 Madrid, Spain; anamorenoblanco1993@gmail.com (A.M.-B.); jrpanno@salud.aragon.es (J.R.P.); 5Microbiology Department, Preventive Medicine and Public Health, University of Zaragoza, 50009 Zaragoza, Spain; 6Institute of Carbochemistry (ICB-CSIC), 50018 Zaragoza, Spain; mpdomingo@icb.csic.es; 7Microbiology Department, University Hospital Ramón y Cajal, Instituto Ramón y Cajal de Investigación Sanitaria (IRYCIS), 28034 Madrid, Spain; rabarb01@ucm.es; 8Infectious Diseases Department, Lozano Blesa University Hospital Clinic, Institute for Health Research Aragón, University of Zaragoza, 50009 Zaragoza, Spain; 9Facultad de Ciencias de la Salud, Universidad Alfonso X El Sabio, 28691 Villanueva de la Cañada, Spain

**Keywords:** NSCLC, predictive biomarkers, ICI response, irAEs, gut microbiota, intestinal permeability, SCFAs, citrulline, zonulin, endotoxin

## Abstract

**Simple Summary:**

There is evidence of the influence of the intestinal microbiota on the response to immunotherapy in cancer. In addition, we lack markers of response to treatment and toxicity, which obliges us to continue our search for them. In this work, we recruited patients with non-small-cell lung cancer receiving immunotherapy who contributed a fecal and blood sample. We analyzed the possible relationship between the response to immune checkpoint inhibitors and the occurrence of immune-related adverse events, the composition (16S rDNA amplification) and functionality (abundance of short-chain fatty acids) of the gut microbiota, and intestinal membrane permeability as a human factor. No correlations were detected between analytical markers and clinical evolution, with a marked individuality of the gut microbiota in terms of composition, but homogeneity in its functionality and permeability.

**Abstract:**

Immune checkpoint inhibitors have been proposed as the standard treatment for different stages of non-small-cell lung cancer in multiple indications. Not all patients benefit from these treatments, however, and certain patients develop immune-related adverse events. Although the search for predictors of response to these drugs is a major field of research, these issues have yet to be resolved. It has been postulated that microbiota could play a relevant role in conditioning the response to cancer treatments; however, the human factor of intestinal permeability also needs to be considered as it is closely related to the regulation of host–microbiota interaction. In this article, we analyzed the possible relationship between the response to immune checkpoint inhibitors and the onset of immune-related adverse events, gut microbiota status, and intestinal membrane permeability. In a pioneering step, we also measured short-chain fatty acid content in feces. Although the correlation analyses failed to identify predictive biomarkers, even when all variables were integrated, our patients’ microbial gut ecosystems were rich and diverse, and the intestinal barrier’s integrity was preserved. These results add new knowledge on the composition of microbiota and its correlation with barrier permeability and short-chain fatty acids and suggest that more studies are required before these potential biomarkers can be incorporated into the clinical management of patients via immune checkpoint inhibitor treatment.

## 1. Introduction

Lung cancer is the fourth most common cancer worldwide, with a high mortality rate. Histologically, lung cancer can be differentiated into small-cell (15–20%) and non-small-cell lung cancer (NSCLC), (80–85%). There have been numerous advances in NSCLC detection, diagnosis, and treatment in the past decade, mainly due to our better understanding of its biological mechanisms. The therapeutic use of immune checkpoint inhibitors (ICIs) significantly improves survival in patients without genetic alterations [1]. However, the overall response rate to ICIs in NSCLC is 15–20%, and no predictive biomarkers of response are available [2]. Furthermore, despite the massive use of ICIs, a considerable number of patients obtain no benefit from them [3], which can lead to immune-related adverse events (irAEs), resulting in reduced quality of life, increased healthcare costs, and severe clinical deterioration and even death.

In addition to the respiratory tract, previous studies have pointed to the gut microbiota’s contribution to both ICI response and toxicity [4]. The gut ecosystem maintains a continuous dialogue with the immune system, which could limit the effectiveness of ICIs due to resource consumption [5]. Nevertheless, the most recent research has focused on microbial metabolites, mainly short-chain fatty acids (SCFAs), which directly induce cell apoptosis [6,7,8]; however, normal values for SCFAs have yet to be established, and only differences between patients and controls or between ICI responders and non-responders have been explored [9].

Intestinal permeability that is closely related to the host response to gut microbiota is another essential human factor in ICI response and typically determined by measuring plasma citrulline, zonulin, and endotoxin levels [10]. Citrulline is an amino acid produced almost exclusively by enterocytes of the small intestine mucosa [11] that can be aberrantly recognized by the immune system, triggering the generation of anticitrulline antibodies. Previous studies have shown a correlation between citrulline levels and ICI response [12,13]. The activation of zonulin, a small protein that modulates the integrity of tight junctions, has been proposed as a defensive mechanism that contributes to the host’s innate immune response against changes in the microbiome ecosystem [14]. Endotoxins, which are part of the outer membrane of Gram-negative bacteria, are highly antigenic and trigger a systemic inflammatory response that can modulate ICI response [15,16].

The aim of the present study was to identify predictive markers of ICI response and toxicity in patients with NSCLC, focusing the search on the intestinal tract and including the gut microbiota and intestinal permeability factors.

## 2. Materials and Methods

### 2.1. Study Design and Patients

This study was approved by the Clinical Research Ethics Committee of Aragón (code C.I. PI19/052), and all patients signed the written consent form. The inclusion criteria were candidates for ICI treatment with stage III or stage IV NSCLC, including de novo diagnosis, or after the resistance to/recurrence of cancer following a treatment other than an ICI. The exclusion criteria were a histology other than NSCLC, a concomitant tumor of another origin, autoimmune disorders contraindicating ICIs, previous immunotherapy, and a score of grade 2 or higher on the Eastern Cooperative Oncology Group (ECOG) scale. Patients undergoing corticosteroid therapy with doses higher than 10 mg/24 h of prednisone or equivalent, as well as those with known immunodeficiencies (including HIV infection), were excluded. Each patient provided baseline blood and stool samples before starting immunotherapy. Samples were collected and managed by the Biobank of Aragón. PDL1 was determined via immunohistochemical staining, classifying the patients into negative (<1%), low expression (1–49%), or high expression (>50%) groups.

The patients’ clinical and anthropometric variables were collected from the clinical chart, and the functional capacity was defined using the validated ECOG scale, with grade 0 being normal capacity and grade 4 being bedridden 100% of the day. The tumor stage was defined according to the grades established by the eighth TNM consensus [tumor (T), node (N), and metastasis (M)] for lung cancer [17]. Tumor response was established according to the Response Evaluation Criteria in Solid Tumors (RECIST) criteria (v.1.1) [18]. The ICI regimen was based on clinical practice guidelines at the time the study was conducted, and clinical follow-up was performed every 2–3 weeks during treatment administration, depending on the type of treatment, and every 2–3 months without therapy.

In patients with locally advanced unresectable stage III, the treatment regimen used was a combination of chemotherapy (carboplatin AUC5 and oral vinorelbine 60 mg/m^2^ days 1 and 8) for 4 cycles, with the last two being concomitantly administered with radiotherapy treatment (54 to 66 Gy); after that, 10 mg/kg of body weight of durvalumab was administered intravenously every 2 weeks for up to 12 months based on the PACIFIC trial [19].

Patients who had previously untreated advanced NSCLC with PD-L1 expression on at least 50% of tumor cells and no sensitizing mutation of the epidermal growth factor receptor gene or translocation of the anaplastic lymphoma kinase gene received pembrolizumab at a fixed dose of 200 mg every 3 weeks as first line of treatment [20].

Nivolumab, provided at a dose of 3 mg/kg weight every 2 weeks, was used in patients with advanced squamous NSCLC who had disease progression during or after first-line chemotherapy [21]. Atezolizumab administered at a dose of 1200 mg every 3 weeks was used in patients who had received one to two previous cytotoxic chemotherapy regimens (one or more platinum-based combination therapies) for stage IIIB or IV NSCLC [22].

### 2.2. Gut Microbiota Characterization

Stool samples were thawed at −20 °C for 24 h and then at 4 °C for another 24 h. Samples were suspended in 500 µL of MiliQ water, and DNA extraction was performed with the QIAamp extraction kit (Quiagen, Hilden, Germany). Using Mi-Seq 2 × 300 bp paired-end technology (Illumina, Inc., San Diego, CA, USA), 16S rDNA massive sequencing was performed on the variable regions V3 and V4 of the 16 rRNA in the Central Translational Genomics Support Unit at the Ramón y Cajal Health Research Institute. Sequences of low quality or small size were eliminated, and representative amplicon sequence variants (ASV) were located and counted. Each ASV was taxonomically assigned to a taxonomic group whose maximum correlation was the genus level (often, several ASVs were assigned to the same group or remained unassigned and had to be removed). A SILVA 138 sequence classifier was employed for taxonomic assignment. Lastly, alpha diversity index (within a sample) was performed using the Shannon index, and the beta-diversity distance between samples was estimated via LEfSe (Linear discriminant analysis Effect Size) [23] using QIIME2 software, version 2021.11.

### 2.3. Short-Chain Fatty Acids in Feces

Aliquots of feces were processed as previously described [24]. In brief, at least 30 mg of feces was extracted with 293.75 µL ethanol, and 6.25 µL of deuterated butyric acid D7 4 g/L as IS mix (CF 25 µg/sample) was added. After vigorous vortexing, the samples were centrifuged for 10 min at 13,000× *g*. The supernatant was transferred to a new Eppendorf tube with 5 µL of freshly prepared 0.8 M sodium hydroxide. Solvents were evaporated using a vacuum centrifuge for 3 h (Thermo Fisher SpeedVac™ SPD121P, Dreieich, Germany). The residual salts were redissolved in 50 µL of ethyl alcohol and acidified with 10 µL of 0.6 M succinic acid immediately before the analysis. Gas chromatography-mass spectroscopy (GC-MS) analysis was performed using a TRACE 1600/1610 gas chromatograph/ISQ7610 mass selective detector (Thermo Fisher Scientific, Dreieich, Germany) equipped with a TG-WaxMS A GC Column (15 m × 0.32 mm × 0.25 μm, Thermo Fisher Scientific, Dreieich, Germany). The injector, GC-MS transfer line, and ion source temperature were set to 200 °C, 215 °C, and 250 °C, respectively. The flow rate of the helium carrier gas initially started at 2.5 mL/min, was maintained for 2 min, and then increased at a rate of 1 mL/min to 5 mL/min, where it was kept for 1 min. We then introduced 1 μL of the sample via splitless injection. The initial column temperature was set to 55 °C and held for 1 min, then raised to 105 °C at a rate of 8 °C/min, where it was held for 2 min. Lastly, the column temperature was raised to 190 °C at a rate of 30 °C/min and kept at this temperature for 1 min. An extra step was added to delete possible leftovers: the temperature was raised to 210 °C at a rate of 20 °C/min and held at this temperature for 3 min. The ionization was performed in the electron impact mode at 70 eV. Initially, the MS data were acquired in full scan mode at a mass-to-charge ratio (*m*/*z*) of 40–130 and with a scan time of 0.2 s. SCFAs were detected between 4.7 min and 12.5 min. The compounds were identified by comparing the obtained MS spectra to the National Institute of Standards and Technology database and confirmed by comparison with the retention times of the pure standards. The instrument was operated, and the data were acquired and analyzed using Chromeleon 7 software.

### 2.4. Serum Factors Related to Intestinal Permeability

Blood was collected in sodium heparin tubes and centrifuged for 10 min at room temperature at 2600 rpm. The cellular fraction was thereby separated from the plasma, which was extracted with a cell pellet and stored at −80 °C for later use. Citrulline quantification was performed via high-performance liquid chromatography at the Institute of Carbochemistry of Zaragoza (ICB-CSIC). As a calibration line, different concentrations of L-citrulline, L-asparagine, and L-arginine were prepared. Sample derivatization was needed for further analysis via fluorescence detection, performed by adding a reconstituted Accq-fluorine. Anticitrulline antibodies were analyzed using chemiluminescence techniques using the BIO-FLASH optical system (Werfen, Barcelona, Spain) in the immunology laboratory of HCU. For quantitative determination in plasma, we employed the Human Zonulin ELISA kit (Cusabio, Catalog Number. CSB-EQ027649HU) with a detection range of 0.625–40 ng/mL. The results were analyzed using the professional software CurveExpert Pro https://www.cusabio.com/c-18069.html (accessed on 10 December 2021). Lastly, plasma endotoxin concentrations were quantitatively analyzed with the ToxinSensor™ Chromogenic LAL Endotoxin Assay Kit (GenScript USA Inc., Piscataway, NJ, USA) according to the manufacturer’s instructions.

### 2.5. Statistical Analysis

The descriptive (clinicopathological variables, treatment and heatmaps) and survival (Kaplan–Meier curves, Cox regression models, and Pearson or Spearman correlations) analyses were performed using Prism 10 and Orange 3 software, considering a *p*-value < 0.05 to be statistically significant.

## 3. Results

Of the 55 patients recruited, 35 (63.6%) were ICI responders, and 25 (45.4%) developed irAEs: 20 of the ICI responders and 5 of the ICI non-responders (Table 1). The most common toxicities were cutaneous and pneumonitis (10.9% each), followed by endocrine toxicity (9.1%). Overall survival was longer for the patients who responded to ICI and for those who developed irAEs, reaching statistical significance for longer progression-free survival in those who had both conditions (Figure 1).

All patients were Caucasian, 39 (70.9%) were male, and 47 of them (85.4%) were over 75 years old. The Eastern Cooperative Oncology Group (ECOG) was zero in 36 patients (65.5%) and one in 19 patients (34.5%). Sixteen patients were stage III (29.1%), and 39 patients were stage IV (70.9%). Twenty-two patients had squamous carcinoma (40%), and the other 33 patients had adenocarcinoma (60%).

The treatment indication was in locally advanced in 14 patients (25.5%), first-line treatment was in 18 patients (32.7%), and second-line treatment or more was in 23 patients (41.8%); moreover, durvalumab was used in 14 patients (25.5), pembrolizumab was used in 21 patients (38.2%), atezolizumab was used in 18 patients (32.7%), and nivolumab was used in 2 patients (2%).

In the 35 patients who had an ICI response, 13 (37.1%) were squamous (9 males), and 22 (62.9%) had adenocarcinoma (15 males). ECOG PS was zero in 28 of them (80%) and one in the other 7 (20%). The mean age was 65 years old. On the other hand, in the 20 patients who did not respond to ICI, 9 (45%) were squamous (8 males) and 11 (55%) had adenocarcinoma (7 males). ECOG PS was zero in 9 of them (45%) and one in the other 11 (55%). The mean age was also 65 years old. Plasmatic LDH was higher in ICI-response, but all patients with a complete ICI response had normal values. LDH values in the 20 patients who did not respond to ICI were above the normal limit.

In the group that developed irAEs (n = 25, 14 males), 12 (48%, 9 males) were squamous and 13 were adenocarcinomas (52%, 7 males). ECOG PS was 0 in 19 of them and 1 in the other 6. The mean age was 68 years old. In the 30 patients (23 males) who did not develop irAEs, 10 were squamous (33%, 8 males) and 20 were adenocarcinomas (67%, 15 males). ECOG PS was 0 in 17 of them and 1 in the remaining patient. The mean age was 63.5 years old.

ECOG PS 1 and squamous histology were more frequent in those patients who did not achieve a response to the ICI treatment. Females tended to have a better response to treatment, and LDH values were normal in all the patients who achieved a complete response to ICI treatment. The mean age was similar in both the responder and non-responder groups. irAE development was similar in both histology subgroups.

In the responder group (n = 35), PDL1 was <1% in 10 (28.5%) patients, 1–49% in 13 (37.4%), and >50% in 11 patients (31.4%). In one patient, the result was not valuable. From the 10 patients who achieved complete response, in 5 of them, it was between 1 and 49%, and in the other 5, it was >50%. On the contrary, in ICI non-responders (n = 20), PDL1 was <1% in five (25%) patients, 1–49% in eight patients (40%), and >50% in five patients (25%), despite the results not being valuable in two patients. PDL1 determination does not correlate with response to ICI, so predictive factors for immunotherapy treatment in NSCLC are urgently needed.

Predictive markers have also been searched in the immune cell profiling [25], concluding that lymphocyte and NK cell populations are of great interest. However, no immune cell profile has yet been identified with sufficient clinical relevance to be established in clinical practice.

### 3.1. Gut Microbiota Composition and Metabolites

The microbiota compositions, considering both alpha and beta diversities, were comparable in all patients regardless of their ICI response or onset of irAEs (Figure 2). However, the LEfSe analysis showed statistically significant differences for the Bacillota phylum, including a higher abundance of *Clostridia* and *Monoglobus* in the ICI responders and *Hungatella* in the ICI non-responders. Moreover, the patients who developed irAEs exhibited a significantly higher abundance of Betaproteobacteria and *Enterococcus*, doing so to the detriment of *Negativicutes* and other genera of the Bacillota phylum. *Akkermansia* was scarcely represented among our patients, being undetectable in almost half of the patients, with few reads in the rest. Correlations with abundance and ICI response or irAEs could, therefore, not be established.

Overall, most SCFAs were acetic (56%), propionic (19%), and butyric (11%) acids, whereas valeric (5%), isovaleric (5%), and isobutyric (4%) acids were detected in lower proportions. The distributions of each SCFA and the subsequent analyses based on ICI response and irAE are shown in Figure 3. No statistically significant differences among the various groups were detected.

### 3.2. Biomarkers in Peripheral Blood

The serological distribution of each parameter’s concentration is shown in Figure 4. Citrulline showed the highest dispersion; the rest of the measurements showed uniform results, especially the anticitrulline antibodies, which were undetectable in 90% of the samples. Lastly, we performed a comprehensive analysis including all clinical and analytical variables (Figure 5); however, no predictive markers for ICI response or irAE onset were identified, and there were no correlations between bacterial abundance, SCFA density, and/or permeability parameters.

## 4. Discussion

Given the current use of ICIs in oncology, with widely diverging results among various types of cancer, research into predictive biomarkers for response and irAE onset is a priority. Despite numerous studies having analyzed various potential biomarkers for predicting responses and irAEs, few of them have shown potential with certain limitations for selecting patients who will best benefit from the therapy [26]. The immunological response is modulated not only by the local tumoral lesion but also by various parts of the body, including the gut (the focus of our study), where local and systemic immune responses are modulated by a close dialogue with the microbiota [27,28]. The various factors of the host gut microenvironment that have been proposed to modulate the immune response against tumors and, specifically, the response to ICIs in lung cancer include the microbiota and, less explored, SCFAs and epithelial barrier markers. Our first approach was to determine the bacterial composition and differential taxa abundance in feces, which has previously been explored by other authors [29,30], with no conclusive results in lung cancer. Alpha diversity values were comparable in all patients and suggested a diverse and rich microbial ecosystem, with median values of 5.5 and 13 for the Shannon and Faith PD indexes, respectively. The LEfSe analysis pointed to specific taxa differentiating ICI responders, a higher abundance of *Clostridia* combined with lower richness of *Hungatella*, and irAE onset with higher amounts of *Betaproteobacteria* to the detriment of *Negativicutes*. Our results agree with previous findings of studies of melanoma and other solid tumors treated with ICIs in which the responders presented enrichment of *Clostridiales* in the gut, compared to an increase in *Hungatella* in non-responders [31]. Derosa et al. and Routy et al. reported a correlation between ICI response and *Akkermansia* abundance [32,33], which was recently confirmed by an independent study [34]. Curiously, this genus was not represented in almost half of our patients and was a minority genus in the rest of the patients. Other studies of melanoma [35] and other solid tumors including lung cancer [27] have shown no correlation between *Akkermansia* and ICI responses.

Most of the previously published studies on microbiota and human disease have sought to identify a specific bacterium as the main contributor; however, these results are often not reproducible in subsequent studies with different patients or conditions. Therefore, despite several decades of studies on microbiota, specific and universal markers for each disease have not been identified.

Bacterial genetics allows for the exchange in metabolic pathways, thereby shifting the focus of research from the composition to the study of the global ecosystem metabolism, rather than the abundance of individual taxa. SCFAs (exclusively bacterial metabolites produced through the fermentation of undigestible fibers) have been implicated in numerous medical conditions, even in NSCLC, although their correlation with ICI response is unknown [36,37]. SCFAs affect intestinal motility, barrier function, host metabolism, and microbiota distribution. Of the SCFAs, butyrate is the most closely related to oncological processes and has been proposed as an inhibitor of cancer cell proliferation, with direct antineoplastic effects due to the interactions with histone deacetylase enzymes [38,39]. Propionate administration has therapeutic potential in reducing NSCLC aggressiveness and warrants further clinical testing [40]. Locally in the lungs, SCFAs might stimulate regulatory T cells, T helper 2 cells, and interleukin 22+ type 3 innate lymphoid cells to protect against airway inflammation via T-cell receptor signaling [41]. Acetic, propionic, and butyric acids have been reported as the main SCFAs in feces, with a distribution of 60:20:20 [42], which is comparable to our results (56:19:11). Despite our robust data analysis, there were no correlations between the SCFAs and bacteria or ICI response, which contrasts with the results of Nomura et al., who observed that an increase in fecal SCFAs correlated with PFS in solid tumors [9]. Standardized methodologies and normal values for SCFAs are not yet available, which hinders comparisons. Our study also focused on lung cancer, in contrast to the study by Nomura et al. [9], in which half of the patients had melanoma and only two had lung cancer. Of note, half of the patients included in the study by Nomura et al. had solid cancers in which ICI efficacy was poor, such as head and neck, gastrointestinal, and genitourinary cancer.

Together with the SCFA analysis, our study’s main contribution is the exploration of intestinal barrier permeability, which has hardly been explored in NSCLC as a host-dependent factor. There is a significant lack of standardized procedures and breakpoint values to determine the permeability, which is a serious limitation of our study. Our results show low dispersion in the four variables (citrulline, anticitrulline antibodies, zonulin, and endotoxin), which rules out the possibility of identifying biomarkers for ICI response and the onset of irAEs. Most previous studies have demonstrated the effect of endotoxins on NSCLC cells in vitro or in animal models. To the best of our knowledge, however, ours is the first report of circulating endotoxin levels in patients with NSCLC differentiated according to ICI response and irAEs onset.

The main limitation of our study is the lack of consensus on the methodological aspects and normal values for the measurements. Nevertheless, we explored the intestinal tract microenvironment as a possible source of biomarkers due to the lack of biomarkers in other compartments [43,44].

## 5. Conclusions

We found no markers related to gastrointestinal tract permeability, microbiome composition, or SCFAs that could contribute to predicting ICI response or irAE onset. Nevertheless, they might help other researchers to continue exploring the intestinal tract as a possible source of biomarkers, as they include features that had not been previously characterized.

## Figures and Tables

**Figure 1 cancers-16-01144-f001:**
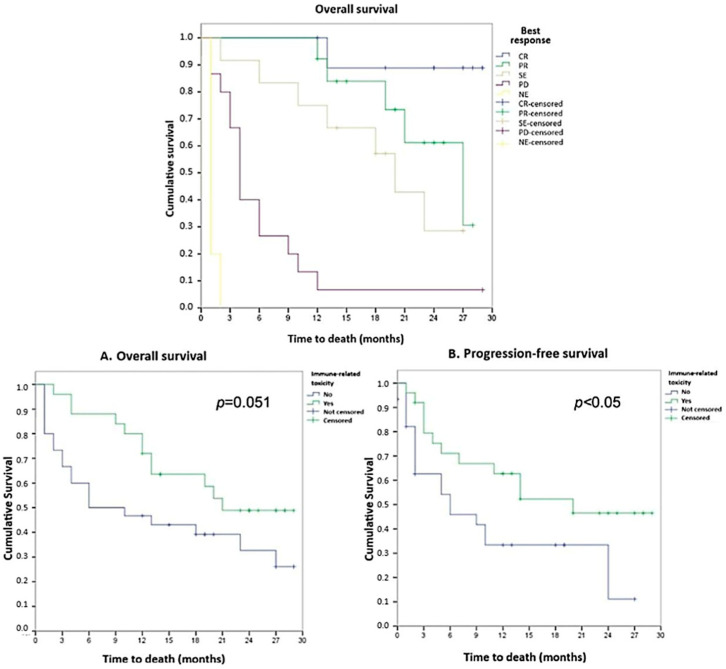
Kaplan–Meier survival analysis according to the presence of immune-related toxicity: (**A**) overall survival; (**B**) progression-free tumor survival. Complete response (CR), partial response (PR), stable disease (SE), progressive disease (PD), and not evaluated (NE).

**Figure 2 cancers-16-01144-f002:**
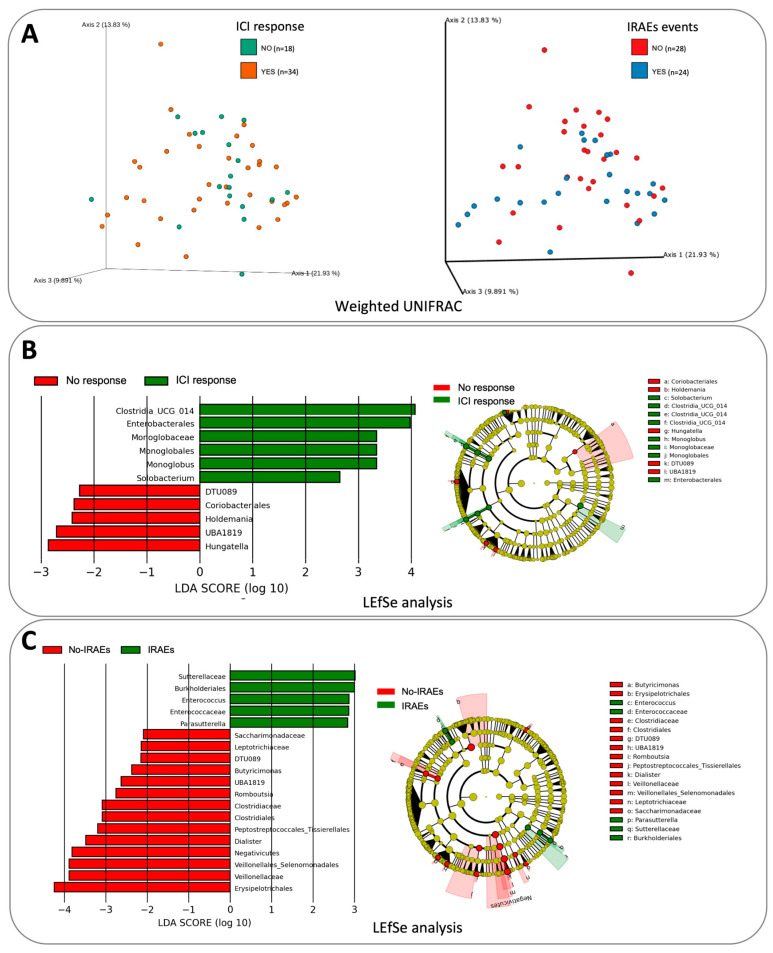
Bacterial composition in the different groups used in this study: (**A**) weighted UNIFRAC analysis that failed to separate responders and non-responders to ICIs, as well as patients with and without irAEs; (**B**) LEFSe results for patients who did and did not respond to ICI treatment; (**C**) LEFSe results for patients who did and did not have ICI toxicity.

**Figure 3 cancers-16-01144-f003:**
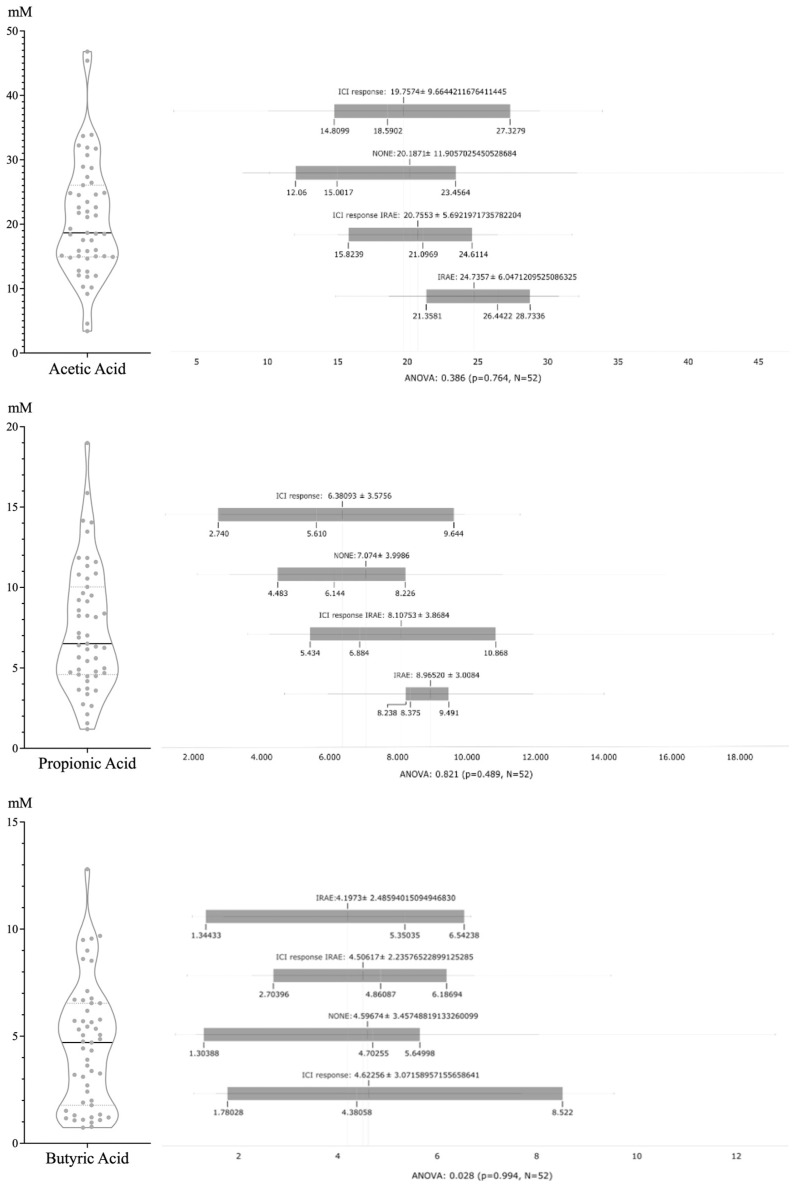
Values obtained for each of the SCFAs and their statistical analysis according to patient group: response, response and toxicity, toxicity only, and none. Statistically significant differences were not detected for the 4 groups or any SCFA.

**Figure 4 cancers-16-01144-f004:**
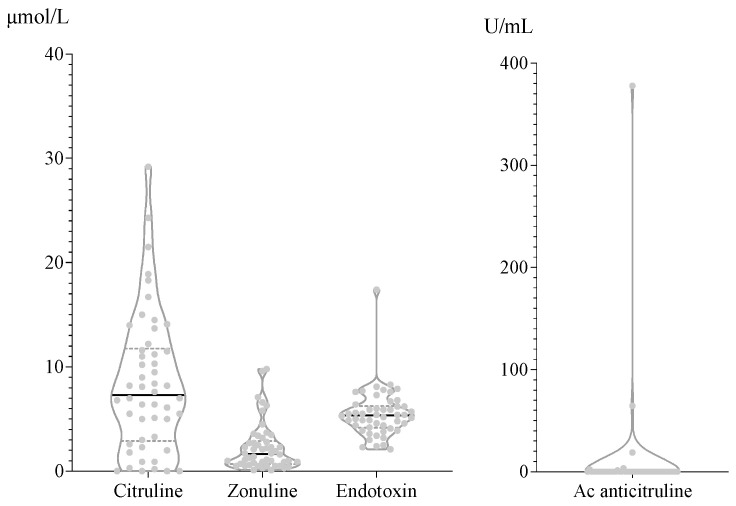
Distribution of values obtained for markers related to intestinal permeability.

**Figure 5 cancers-16-01144-f005:**
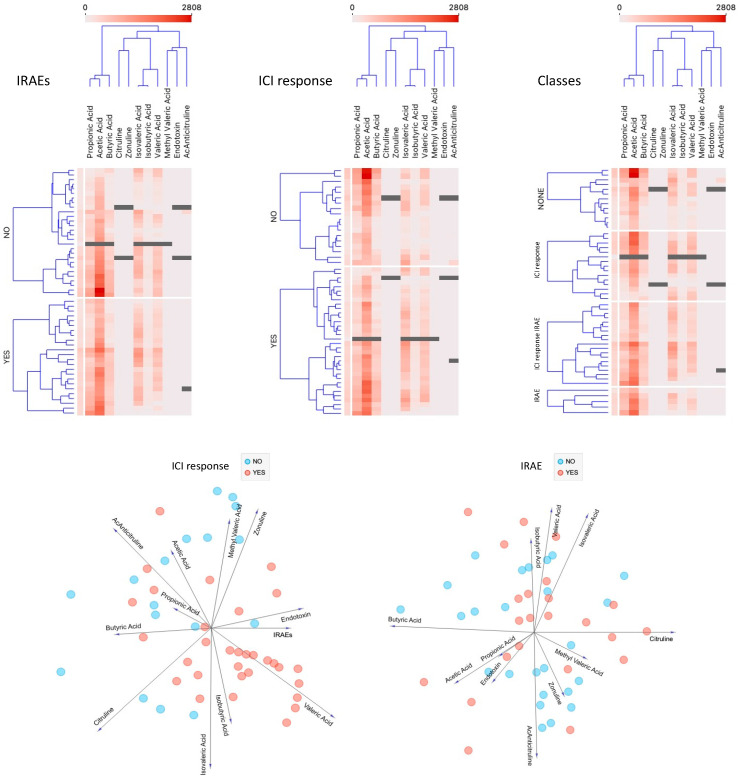
Integrative statistical analysis including all clinical and analytical variables.

**Table 1 cancers-16-01144-t001:** Patients baseline characteristics.

Characteristics	N = 55	%
Sex		
Male	39	70.9%
Female	16	29.1%
Age		
<75	47	85%
≥75	8	15%
ECOG		
ECOG 0	36	65.5%
ECOG 1	19	34.5%
Histology		
Squamous	22	40%
Adenocarcinoma	33	60%
Tumor stage		
Stage III	16	29.1%
Stage IV	39	70.9%
Treatment indication		
Locally advanced	14	25.5%
First line	18	32.7%
Second line or more	23	41.8%
ICI drug		
Durvalumab	14	25.5%
Pembrolizumab	21	38.2%
Atezolizumab	18	32.7%
Nivolumab	2	2%

## Data Availability

The datasets generated and analyzed in the current study are available from the corresponding author upon reasonable request.

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
