# Peer review of "Biomarkers of Immunotherapy Response in Patients with Non-Small-Cell Lung Cancer: Microbiota Composition, Short-Chain Fatty Acids, and Intestinal Permeability"

_cancers, 2024, doi:10.3390/cancers16061144_

Round 1
Reviewer 1 Report
Comments and Suggestions for Authors
The manuscript explores predictive biomarkers for immune checkpoint inhibitor (ICI) response and immune-related adverse events (irAEs) in non-small cell lung cancer (NSCLC) patients. The study focuses on the gut microbiota, short-chain fatty acids (SCFAs), and intestinal permeability. Out of 55 patients, 35 (63.6%) were ICI responders, and 25 (45.4%) experienced irAEs with common toxicities included cutaneous and pneumonitis. They explored gut microbiota composition and SCFA distribution, but no significant differences were found among groups. Additionally, serum factors related to intestinal permeability showed low dispersion, hindering biomarker identification. Overall, they conclude that no identified markers in gastrointestinal permeability, microbiome composition, or SCFAs contribute to predicting ICI response or irAE onset and encourages further exploration of the intestinal tract as a potential source of biomarkers.
While the manuscript provides a thorough and impactful contribution to the understanding of predictive biomarkers for ICI response and irAEs in NSCLC., the following topics could be considered for elaboration:
1. Detailed demographic information, including age, gender, ethnicity, and relevant medical history, should be included in responders and non-responders to provide a comprehensive understanding of the study population.
2. Conduct a more extensive correlation analysis, exploring relationships between gut microbiota, SCFAs, and intestinal permeability, as well as their correlations with ICI response and irAEs. These could be included as follows:
1) Perform subgroup analyses based on different NSCLC subtypes, stages, or other relevant clinical characteristics to uncover potential variations in the observed relationships.
2) Conduct functional analysis of the gut microbiota to gain insights into specific microbial functions that may influence ICI response and irAEs.
3) Predict metabolic pathways based on the microbial community composition. Investigate the potential metabolic contributions of the gut microbiota to SCFA production.
4) Stratify patients based on specific microbial signatures and investigate whether certain microbial profiles are associated with better or worse ICI response or irAE outcomes.
5) Conduct phylogenetic analysis to infer the evolutionary relationships among microbial taxa. Explore whether phylogenetic diversity is associated with treatment outcomes.
6) Conduct a more detailed analysis of SCFAs, including exploring their individual roles, potential interactions, and specific effects on the tumor or tumor microenvironment.
7) Collect and analyze additional clinical parameters that may impact ICI response and irAEs, such as genetic markers, immune cell profiling, or inflammatory markers.
Author Response
The authors of this paper thank you for your review and questions that help us to improve the quality of our manuscript. All questions have been responded one by one.
- Detailed demographic information, including age, gender, ethnicity, and relevant medical history, should be included in responders and non-responders to provide a comprehensive understanding of the study population.
Response: It has been included in the results section.
- Conduct a more extensive correlation analysis, exploring relationships between gut microbiota, SCFAs, and intestinal permeability, as well as their correlations with ICI response and irAEs. These could be included as follows:
Response: All these analyses have been performed with no positive results. We have contacted the data science unit of our center for a thorough analysis, but unfortunately it has been unsuccessful.
- Perform subgroup analyses based on different NSCLC subtypes, stages, or other relevant clinical characteristics to uncover potential variations in the observed relationships.
Response: Done with negative results
- Conduct functional analysis of the gut microbiota to gain insights into specific microbial functions that may influence ICI response and irAEs.
Response: It has been done.
- Predict metabolic pathways based on the microbial community composition. Investigate the potential metabolic contributions of the gut microbiota to SCFA production.
Response: Special attention has been paid to the abundance of bacterial species and to the abundance of SCFA, with no positive results. Although other authors make a functional prediction with the composition based on 16S rDNA, we believe that this is a bit risky, since we cannot attribute a universal functionality to the genus but is specific to each genetic lineage.
- Stratify patients based on specific microbial signatures and investigate whether certain microbial profiles are associated with better or worse ICI response or irAE outcomes.
Response: It has been done.
- Conduct phylogenetic analysis to infer the evolutionary relationships among microbial taxa. Explore whether phylogenetic diversity is associated with treatment outcomes.
Response: Phylogenetic analysis with 16S rDNA is complicated, for that we should do shotgun, but our center does not have the bioinformatics processing for this technique.
- Conduct a more detailed analysis of SCFAs, including exploring their individual roles, potential interactions, and specific effects on the tumor or tumor microenvironment.
Response: It has been done without relevant results.
7) Collect and analyze additional clinical parameters that may impact ICI response and irAEs, such as genetic markers, immune cell profiling, or inflammatory markers.
Response: It has been done and included in the new versión.
Reviewer 2 Report
Comments and Suggestions for Authors
My problem with the whole article is that I don't see the hypothetical molecular link between gut microbiome-SCFA and the therapeutic response of ICI-treated NSCLC patients that the authors were looking for. And this is acknowledged by the authors.
The article was written using completely novel, cutting-edge technologies and statistical methods, so I consider it technically acceptable. But there is still a hiatus about the basic idea.
I suggest that the authors dig deeper into exploring possible correlations.
A few suggestions:
How many patients had ICI colitis?
What were the changes in IL25 levels in the patients? Were DCLK1-positive tuft cells detected in the colon? For example, the microbiome-SCFA-tuft cell-IL25 linkage could represent a hypothetical pathway to NSCLC.
Author Response
The authors of this paper thank you for your review and questions that help us to improve the quality of our manuscript.
The lack of biomarkers of response to immunotherapy in lung cancer forces us to continue research. The idea of searching for the composition and functionality of the intestinal microbiota arises from the studies in which it has been described that a fecal transplant can modulate the response to ICI. See review: (Kiousi et al The Role of the Gut Microbiome in Cancer Immunotherapy: Current Knowledge and Future Directions. Cancers (Basel). 2023 Mar 31;15(7):2101. doi: 10.3390/cancers15072101 and Zhang H, Xu Z. Gut-lung axis: role of the gut microbiota in non-small cell lung cancer immunotherapy. Front Oncol. 2023 Nov 22;13:1257515. doi: 10.3389/fonc.2023.1257515).
Our patients had no gastrointestinal symptomatology, so we can rule out the existence of any inflammatory disease that had not yet been diagnosed, but we cannot have data on IL-25 non-DCLK1-positive tuft cells in colon.
Reviewer 3 Report
Comments and Suggestions for Authors
Manuscript cancers-2899363
„Biomarkers of Immunotherapy Response in Patients with Non-Small-Cell Lung Cancer: Microbiota Composition, Short-Chain Fatty Acids, and Intestinal Permeability” for Cancers
Comments:
1. Materials and methods. 2.1. Could Authors indicate exactly the trials of ICIs and treatment regimen used in studies for selected stages of lung cancers?
2. Figure 2, If possible, please enlarge in legend the font detailing the detected bacteria.
Author Response
The authors of this paper thank you for your review and questions that help us to improve the quality of our manuscript.
ICIs treatment and their reference studies have been included in the new version.
The resolution of Figure 2 was optimized.
Round 2
Reviewer 2 Report
Comments and Suggestions for Authors
The corrections, additions, and explanations made by the authors now shed light on the problem raised earlier and the interrelationships between the factors of the study.
The revised version of the manuscript is now acceptable for publication.